# WRN and WRNIP1 ATPases impose high fidelity on translesion synthesis by Y-family DNA polymerases

**Jung Hoon Yoon, Karthi Sellamuthu, Louise Prakash, Satya Prakash***

Department of Biochemistry and Molecular Biology, University of Texas Medical Branch at Galveston, Galveston, United States

## eLife Assessment

This manuscript reports an **important** finding for understanding the molecular mechanisms of mutagenesis, carcinogenesis, and senescence. It follows a previous report showing that the Werner syndrome protein WRN and its interacting protein WRNIP1 are indispensable for translesion DNA synthesis (TLS) by Y-family DNA polymerases (Pols). The manuscript provides **convincing** evidence that WRN and WRNIP1 ATPases, in addition to the previously reported role of the WRN 3'>5' exonuclease activity, are essential for promoting the fidelity of replication through DNA lesions by Y-family Pols in human cells.

*For correspondence:
saprakas@utmb.edu

**Abstract** Y-family DNA polymerases (Pols) are intrinsically highly error-prone; yet they conduct predominantly error-free translesion synthesis (TLS) in normal human cells. In response to DNA damage, Y-family Pols assemble and function together with WRN, WRNIP1, and Rev1 in TLS. Among these proteins, WRN possesses a 3'→5' exonuclease activity and an ATPase/3'→5' DNA helicase activity, and WRNIP1 has a DNA-dependent ATPase activity. In a previous study, we identified a role of WRN 3'→5' exonuclease activity in the high in vivo fidelity of TLS by Y-family Pols. Here we provide evidence for a crucial role of WRN and WRNIP1 ATPase activities in raising the fidelity of TLS by these Pols. Defects in WRN and WRNIP1 ATPase activities cause a diversity of nucleotide (nt) misincorporations opposite DNA lesions by Y-family Pols, implicating an unprecedented role of these activities in restraining nt misincorporations, which they could accomplish by tightening the active site of the TLS Pol. Altogether, the combined actions of WRN and WRNIP1 ATPases in preventing nt misincorporations and of WRN exonuclease in removing misinserted nts confer such an enormous rise in the fidelity of Y-family Pols that they perform error-free TLS – essential for genome stability and cellular homeostasis.

## Introduction

Translesion synthesis (TLS) DNA polymerases (Pols) promote replication through DNA lesions. Among the TLS Pols, Y-family Pols play an eminent role in conducting replication through a diversity of DNA lesions. Unlike the replicative Pols which have a constrained active site and synthesize DNA with a high fidelity, Y-family Pols have a more open active site; consequently, they synthesize DNA opposite DNA lesions with an extremely low fidelity. For example, purified Pol $\eta$ replicates through UV-induced cyclobutane pyrimidine dimers (CPDs) efficiently by its ability to accommodate two template nucleotides (nts) in its active site; but because of its open active site, it misinserts nts opposite the 3'T or the 5'T of a *cis-syn* TT dimer with a very high frequency of ~$10^{-2}$ (*Johnson et al., 1999b*; *Masutani et al., 1999*; *Johnson et al., 2000*; *Biertümpfel et al., 2010*; *Silverstein et al., 2010*). Yet, in human or

mouse cells, Pol $\eta$ conducts error-free TLS through CPDs (*Yoon et al., 2009*; *Yoon et al., 2019b*), and mutational inactivation of Pol $\eta$ results in the skin cancer-prone syndrome xeroderma pigmentosum variant (XPV; *Johnson et al., 1999a*; *Masutani et al., 1999*). Overall, TLS studies opposite a variety of DNA lesions have indicated that in spite of their intrinsic high error-proneness, TLS by Y-family Pols operates in a predominantly error-free manner in human cells (not derived from cancers) (*Nair et al., 2006*; *Conde et al., 2015*; *Yoon et al., 2017*; *Yoon et al., 2018*; *Yoon et al., 2019a*; *Yoon et al., 2021a*).

To explain the vast discrepancy between low-fidelity synthesis in vitro vs. predominantly error-free TLS in vivo, we hypothesized that in vivo, TLS Pols associate with other protein factors forming a multiprotein ensemble and that components of the ensemble raise the fidelity of the TLS Pol (*Nair et al., 2006*; *Yoon et al., 2009*; *Conde et al., 2015*; *Yoon et al., 2017*; *Yoon et al., 2018*; *Yoon et al., 2019a*; *Yoon et al., 2019b*; *Yoon et al., 2021a*). To this end, in a previous study, we showed that in response to DNA damage, Werner syndrome protein WRN, WRN interacting protein WRNIP1, and Rev1 assemble together with Pol $\eta$, Pol $\iota$, or Polκ, and that they are indispensable for TLS by Y-family Pols (*Yoon et al., 2024*). Rev1 is a member of the Y-family Pols. However, opposite a number of DNA lesions including UV lesions, Rev1's polymerase activity is not required; instead, it functions as a scaffolding protein for the assembly of the other protein components with the Y-family Pols (*Yoon et al., 2015*). WRN is a member of the Rec Q family of DNA helicases (*Chu and Hickson, 2009*). It possesses two enzymatic activities: a 3'→5' exonuclease activity in the amino-terminal region (*Huang et al., 1998*; *Kamath-Loeb et al., 1998*; *Huang et al., 2000*; *Perry et al., 2006*) and an ATPase/3'→5' DNA helicase activity in the central region (*Gray et al., 1997*; *Shen et al., 1998*). WRNIP1 is a member of the AAA +ATPase family and harbors a DNA-dependent ATPase activity (*Tsurimoto et al., 2005*). Pull-down studies have indicated evidence of direct physical interaction of WRN with WRNIP1 and Pol $\eta$ (*Kawabe et al., 2001*; *Kawabe et al., 2006*; *Maddukuri et al., 2012*).

Previously, we identified a role of WRN's 3'→5' exonuclease activity in the high in vivo fidelity of TLS by Y-family Pols (*Yoon et al., 2024*). In particular, we showed that defects in WRN's exonuclease activity confer error-proneness on error-free TLS through CPDs by Pol $\eta$, and on error-free TLS opposite 1,$N^6$ ethenodeoxyadenosine (εdA) by Pol $\iota$ (*Yoon et al., 2024*). The εdA adduct is formed by interaction of DNA with aldehydes derived from lipid peroxidation in cell membranes. Defects in WRN exonuclease activity additionally confer a loss in fidelity of error-free TLS opposite the oxidative DNA lesion thymine glycol (Tg) by Polκ and elevate the error-proneness of Pol $\eta$ and Pol $\iota$ dependent TLS opposite UV-induced (6-4) photoproducts (PPs). Thus, by removing nts misinserted opposite DNA lesions by the Y-family Pols, WRN's 3'→5' exonuclease activity improves the fidelity of TLS by these Pols (*Yoon et al., 2024*).

Even though WRN exonuclease raises the fidelity of TLS by Y-family Pols, that alone does not account for the vast increase in the in vivo TLS fidelity of these Pols over that of purified TLS Pols. For example, the prevalence of only the C>T or CC >TT mutational hot spots in UV-induced mutational spectra generated opposite CPDs by Pol $\eta$ in WRN exonuclease-deficient cells (*Yoon et al., 2024*) would imply that Pol $\eta$-mediated TLS opposite the T residue of a CPD formed at the CT, TC, or TT dipyrimidine sequence occurs in a predominantly error-free manner. However, that seems highly unlikely, in view of the fact that purified Pol $\eta$ misinserts different nts opposite both the 3' and 5' T residues of a *cis-syn* TT dimer (*Johnson et al., 2000*). Considerations such as this raised the possibility that the fidelity of TLS by Y-family Pols is elevated by additional means.

In extensive biochemical studies, WRN helicase has been shown to unwind a variety of DNA structures including bubbles, D-loops, triplexes, and G-quartets (*Chu and Hickson, 2009*). A role of WRN helicase activity in the replication of the G-rich telomeric strand prevents telomere loss from individual sister chromatids and averts chromosomal fusions (*Crabbe et al., 2004*; *Crabbe et al., 2007*). And WRN helicase activity is involved in maintenance of common fragile site stability (*Pirzio et al., 2008*).

The role of WRN's helicase activity in the unwinding of DNA containing secondary structures is well established; nevertheless, we hypothesized that in the context of the Y-family Pol ensemble comprised of WRN, WRNIP1, and Rev1, with either Pol $\eta$, Pol $\iota$, or Polκ, WRN's ATPase activity may function in an entirely different manner wherein it modulates the fidelity of the TLS Pol. Likewise, in the context of Y-family Pol ensemble, WRNIP1 ATPase activity could similarly impact the fidelity of the TLS Pol. In accord with this, here we provide evidence for a crucial role of WRN and WRNIP1 ATPase activities in raising the fidelity of TLS by Y-family Pols. Thus, the fidelity of TLS by the Y-family Pols is raised by the

combined actions of WRN and WRNIP1 ATPase activities and the WRN exonuclease activity, such that intrinsically highly error-prone TLS Pols perform error-free TLS, and thereby protect against genome instability and tumorigenesis.

## Results

### Proficiency of replication through UV lesions is not affected by defects in WRN and WRNIP1 ATPase activities

Even though WRN DNA helicase activity is not required for TLS opposite a *cis*-syn TT dimer or a (6-4) TT photoproduct carried on the leading strand template in a duplex plasmid (*Yoon et al., 2024*), the proficiency of WRN helicase for unwinding DNA containing secondary structures raised the possibility that this activity might increase the proficiency of Y-family Pols for replicating through DNA lesions in the genomic context. To check for this, we monitored replication fork (RF) progression through UV lesions on single DNA fibers in WRN$^{-/-}$ HFs carrying the vector or expressing either wild-type WRN or ATPase-defective K577A WRN (*Figure 1—figure supplement 1*). And to determine the effects of combinations of WRN ATPase/helicase, WRNIP1 ATPase, and WRN 3′ →5′ exonuclease activities on RF progression through UV lesions, we extended these studies to WRNIP1 depleted WRN$^{-/-}$ HFs expressing both the ATPase-defective K577A WRN and K274A WRNIP1 proteins or expressing K274A WRNIP1 together with the 3′→5′ exonuclease and helicase-defective E84A, K577A WRN (*Figure 1—figure supplement 1*). The E84A WRN mutation has been described previously (*Yoon et al., 2024*). The K577A WRN and K274A WRNIP1 mutations are in the conserved Walker A motif (*Figure 1—figure supplement 1*) involved in ATP binding (*Kawabe et al., 2001*; *Kawabe et al., 2006*; *Newman et al., 2021*).

HFs were pulse-labeled with iododeoxyuridine (IdU) for 20 min, then UV irradiated (10 J/m$^2$) followed by labeling with chlorodeoxyuridine (CldU) for 20 min (*Figure 1A*). Since RF progression through UV lesions occurs as proficiently in HFs defective in WRN ATPase or defective in both the WRN and WRNIP1 ATPase activities as in WT cells, both these activities have no perceptible effect on the proficiency of replication through UV lesions by Y-family Pols (*Figure 1*). Moreover, RF progression through UV lesions was not affected in cells lacking the WRN exonuclease activity as well as the WRN and WRNIP1 ATPase activities (*Figure 1*). Accordingly, the accumulation of WRN or WRNIP1 into UV-induced replication foci was not affected by the K577A and K274A mutations in these proteins (*Figure 1—figure supplement 2*).

### Defects in WRN ATPase, WRNIP1 ATPase, and WRN 3′→5′ exonuclease confer an immense increase in error-proneness upon error-free TLS through CPDs by Polη

TLS through CPDs is conducted by a Pol$\eta$ dependent error-free pathway or by Polθ/Polκ and Polθ/Pol$\zeta$ dependent error-prone pathways (*Yoon et al., 2019b*). Although WRN and WRNIP1 are required for TLS opposite CPDs by Pol$\eta$ and Polκ, the WRN exonuclease activity functions only in removing Pol$\eta$ errors (*Yoon et al., 2024*). In the last section of studies described below, we confirm that defects in WRN and WRNIP1 ATPase activities also have no effect on the fidelity of Polκ; hence, the data described below result from the effects of these activities on the fidelity of TLS opposite CPDs by Pol$\eta$.

To determine the effects of WRN or WRNIP1 ATPase activities on the fidelity of Pol$\eta$ for TLS through CPDs, we analyzed the effects of mutational inactivation of these activities on the frequency of UV-induced mutations resulting from TLS through CPDs in the *cII* gene which has been integrated into the genome of big blue mouse embryonic fibroblasts (BBMEFs). The spectrum of mutations induced by UV and other DNA damaging agents in the *cII* gene resembles that determined from sequence analyses of endogenous chromosomal genes and from whole genome sequence analysis (*You et al., 2001a*; *You and Pfeifer, 2001b*; *Besaratinia and Pfeifer, 2006*; *Alexandrov et al., 2013*; *Martincorena et al., 2015*). To examine UV mutations that result from TLS specifically through CPDs, (6-4) photoproducts are selectively removed by expressing a (6-4) PP photolyase in the BBMEF cell line and treating with photoreactivating light. In WRN-depleted cells expressing WT WRN, spontaneous mutations occur with a frequency of ~17 x 10$^{-5}$ (*Table 1*), whereas in UV-irradiated cells exposed to photoreactivating light to effect (6-4) PP removal by the (6-4) PP photolyase, the mutation frequency

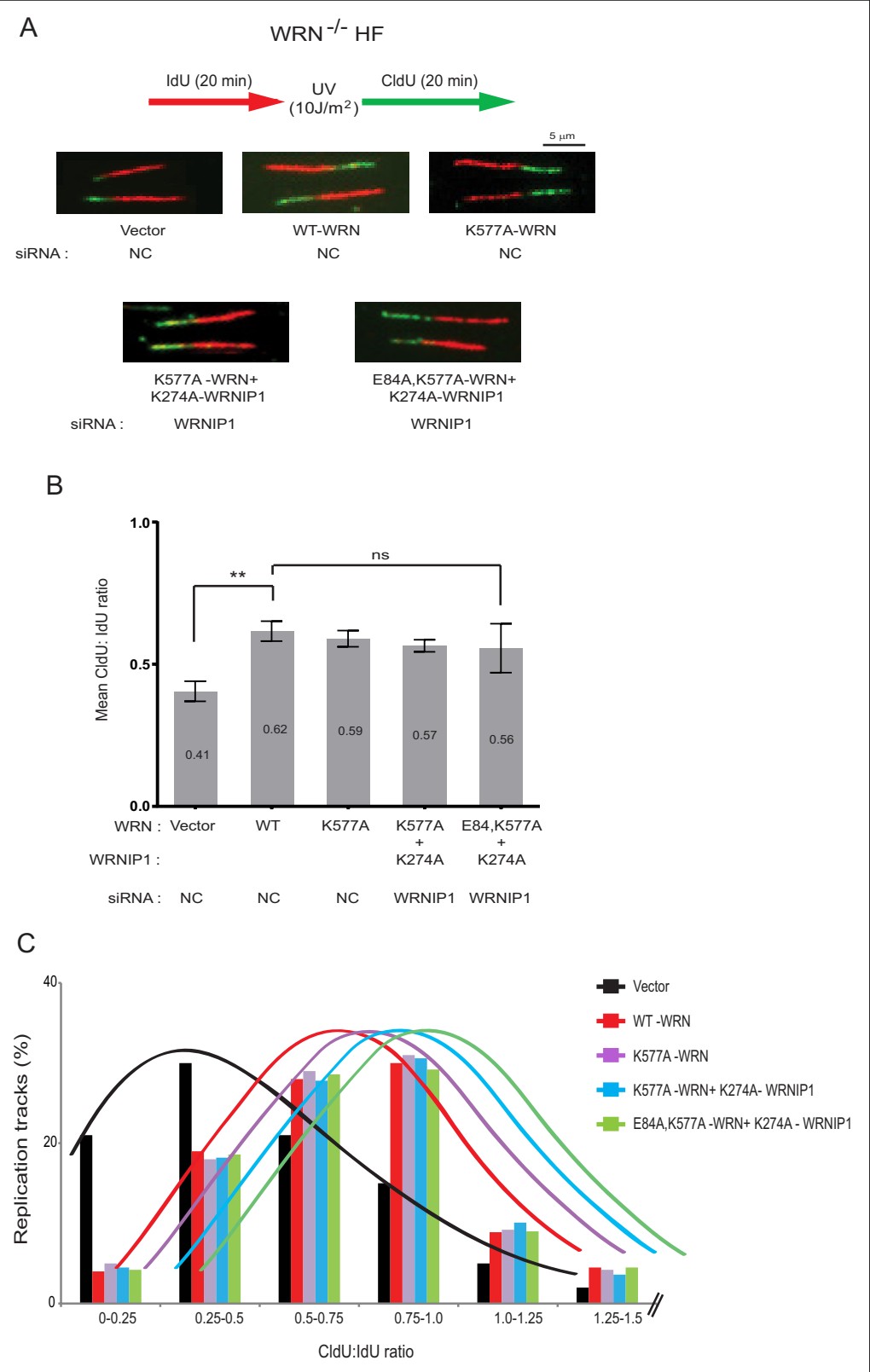

**Figure 1.** Defects in WRN or WRNIP1 ATPase activity do not impair RF progression through UV lesions.
(**A**) Schematic of DNA fiber assay and representative images of stretched DNA fibers in UV-irradiated WRN[-/-] HFs expressing WT WRN or mutant WRN/WRNIP1 proteins. (**B**) Quantitative analyses of RF progression through UV lesions represented as mean CldU/IdU ratio based on ~400 DNA fibers from four independent experiments. Error

*Figure 1 continued on next page*

*Figure 1 continued*

bars indicate SD. Student's two-tailed t test p values: ns, not significant; ** p<0.01. (**C**) Distribution of CldU/IdU ratios in % of replication tracts measured in WRN$^{-/-}$ HFs expressing WT WRN or mutant WRN/WRNIP1 proteins. The mean CldU/IdU ratios for these data are shown in (**B**).

The online version of this article includes the following source data and figure supplement(s) for figure 1:

**Source data 1.** Quantification of CldU/IdU ratio in individual DNA fibers for the data show in *Figure 1B*.

**Figure supplement 1.** siRNA knockdown efficiency of WRNIP1 and stable expression of WT and mutant WRN and WRNIP1 proteins.

**Figure supplement 1—source data 1.** Original uncropped images for western blots shown in *Figure 1—figure supplement 1*.

**Figure supplement 1—source data 2.** Original uncropped images for western blots shown in *Figure 1—figure supplement 1* (labelled).

**Figure supplement 2.** Accumulation of K274A WRNIP1 or K577A WRN into UV-induced replication foci.

---

rises to ~47 x 10$^{-5}$ (*Table 1*). Thus, the UV-induced mutation frequency resulting from TLS opposite CPDs in WT cells is ~30 x 10$^{-5}$ (*Table 1*, last column; *Figure 2A*). As shown previously (*Yoon et al., 2019b*), the entirety of UV-induced mutations in WT cells derives from the Polθ-dependent pathway of error-prone TLS opposite CPDs. In UV-irradiated cells expressing the 3'→5' exonuclease-defective E84A WRN, the UV-induced mutation frequency rises to ~56 x 10$^{-5}$ (*Table 1*, last column), indicating that the E84A mutation confers an ~27 x 10$^{-5}$ increase in the UV-induced mutation frequency over WT cells (*Figure 2A*). Expression of the ATPase-defective K577A WRN increased the UV-induced mutation frequency to ~61 x 10$^{-5}$ and expression of the combined E84A, K577A WRN increased the UV-induced mutation frequency to ~86 x 10$^{-5}$ (*Table 1*, last column). Thus, a defect in WRN's ATPase activity confers an increase of ~31 x 10$^{-5}$ in UV-induced mutation frequency over that in WT cells, and a deficiency in both the WRN exonuclease and ATPase activities confers an additive increase of ~56 x 10$^{-5}$ in UV-induced mutation frequency over that in WT cells (*Figure 2A*).

Next, we determined the effects of the ATPase-defective K274A WRNIP1 mutation on the frequency of UV-induced mutations in the *cII* gene resulting from TLS through CPDs. In cells expressing WT WRNIP1, UV-induced mutations occur at a frequency of ~29 x 10$^{-5}$, and this frequency rises to ~62 x 10$^{-5}$ in cells expressing K274A WRNIP1 (*Table 1*, last column). The increase of ~33 x 10$^{-5}$ in mutation frequency over that in WT cells represents the error-proneness conferred upon Pol$\eta$ TLS by the K274A WRNIP1 mutation (*Table 1*, *Figure 2A*).

We then examined the effects of combinations of these WRN and WRNIP1 mutations on the frequency of UV-induced mutations. In cells expressing the combination of E84A WRN and K274A WRNIP1, or expressing the combination of K577A WRN and K274A WRNIP1, UV-induced mutation frequency rises to ~86 x 10$^{-5}$ and 89 x 10$^{-5}$, respectively, and in BBMEFs expressing E84A, K577A WRN together with K274A WRNIP1, UV-induced mutation frequency rises to ~119 x 10$^{-5}$ (*Table 1*, last column). Thus, the combination of defects in WRN 3'→5' exonuclease and in both the WRN and WRNIP1 ATPase activities confers vast error-proneness on error-free TLS through CPDs by Pol$\eta$, such that Pol$\eta$ generates mutations at a frequency of ~90 x 10$^{-5}$ (*Figure 2A*, last bar).

Since Polθ-dependent TLS through CPDs requires Polκ or Pol$\zeta$ for the extension of synthesis from the nt inserted opposite the CPD by Polθ (*Yoon et al., 2019b*), the possibility remained that in addition to elevating the error-proneness of Pol$\eta$, defects in WRN or WRNIP1 ATPase raise the error-proneness of Polκ; in that case, the increase in mutation frequency would have resulted from an increase in the error-proneness of both Pol$\eta$ and Polκ and not just Pol$\eta$. Hence, since in the absence of Polθ only Pol$\eta$ mediated TLS would be active, we analyzed the effects of E84A WRN and K577A WRN on UV-induced mutations in BBMEFs co-depleted for WRN and Polθ and the effects of K274A WRNIP1 in BBMEFs co-depleted for WRNIP1 and Polθ. Our results that in Polθ-depleted cells, the E84A WRN, K577A WRN, E84A K577A WRN, and K274A WRNIP1 mutations confer nearly the same level of increase in UV-induced mutation frequencies (*Table 1*, last column; *Figure 2B*) as that in Polθ-proficient cells (*Table 1*, *Figure 2A*) establish that defects in the WRN 3'→5' exonuclease activity, WRN ATPase activity, and the WRNIP1 ATPase activity impose immense error-proneness on error-free TLS through CPDs by Pol$\eta$.

**Table 1.** UV-induced mutation frequencies resulting from TLS through CPDs in the *cII* gene in BBMEFs expressing ATPase-defective K577A WRN, 3'→5' exonuclease-defective E84A WRN, ATPase-defective K274A WRNIP1, or combinations of these mutant proteins.

| siRNA/ART558 | Vector expressing | UV* | Photo-reactivation† | Mutation frequency (x10$^{-5}$)‡ | UV induced mutation frequency (x10$^{-5}$) |
|---|---|---|---|---|---|
| WRN | Myc-WT-WRN | - | + | 17.3±0.8 | - |
| WRN | Myc-WT-WRN | + | + | 46.9±2.4 | 29.6§ |
| WRN | Myc-E84A-WRN | + | + | 73.7±3.0 | 56.4 |
| WRN | Myc-K577A-WRN | + | + | 78.2±1.4 | 60.9 |
| WRN | Myc-E84A,K577A-WRN | + | + | 103.3±2.9 | 86 |
| WRNIP1 | Flag-WT-WRNIP1 | + | + | 46.0±1.2 | 28.7 |
| WRNIP1 | Flag-K274A-WRNIP1 | + | + | 79.4±2.7 | 62.1 |
| WRN +WRNIP1 | Myc-E84A-WRN+Flag-K274A-WRNIP1 | + | + | 102.9±4.9 | 85.6 |
| WRN +WRNIP1 | Myc-K577A-WRN+Flag-K274A-WRNIP1 | + | + | 106.0±1.8 | 88.7 |
| WRN +WRNIP1 | Myc-E84A,K577A-WRN+Flag-K274A-WRNIP1 | + | + | 136.3±2.5 | 119 |
| WRN +Pol θ | Myc-WT-WRN | + | + | 19.9±1.2 | - |
| WRN +Pol θ | Myc-E84A-WRN | + | + | 49.7±1.6 | 29.8¶ |
| WRN +Pol θ | Myc-K577A-WRN | + | + | 54.2±1.6 | 34.3 |
| WRN +Pol θ | Myc-E84A,K577A-WRN | + | + | 78.3±2.1 | 58.4 |
| WRNIP1+Pol θ | Flag-WT-WRNIP1 | + | + | 20.6±1.4 | - |
| WRNIP1+Pol θ | Flag-K274A-WRNIP1 | + | + | 60.1±2.7 | 39.5¶ |
| ART558 (20 µM) | - | - | + | 20.6±0.7 | - |
| WRN +WRNIP1/ART558 (20 µM) | Myc-E84A-WRN+Flag-K274A-WRNIP1 | + | + | 77.2±1.2 | 56.6¶ |
| WRN +WRNIP1/ART558 (20 µM) | Myc-K577A-WRN+Flag-K274A-WRNIP1 | + | + | 78.9±1.6 | 58.3 |

*5 J/m$^2$ of UVC (254nm) light.

†Photoreactivation with UVA (360nm) light for 3 hr in cells expressing (6-4)PP photolyase.

‡Data are represented as mean ± SEM. Mean mutation frequencies and standard error of the mean were calculated from 3-4 independent experiments.

§UV-induced mutation frequencies were calculated by subtracting the spontaneous mutation frequency in unirradiated cells (17.3 x 10$^{-5}$) from the mutation frequency in UV-irradiated cells.

¶UV-induced mutation frequencies were calculated by subtracting the basal level of mutations in WT cells depleted or inhibited for Pol θ. Since error-prone TLS by Pol θ is inactivated, these mutations represent error-proneness imposed upon Pol η by the inactivation of WRN and/or WRNIP1 activities.

The online version of this article includes the following source data for table 1:

**Source data 1.** Mutation frequencies from independent experiments for the data shown in *Table 1*.

Next, we verified the effects of combinations of E84A WRN and K274A WRNIP1 or K577A WRN and K274A WRNIP1 in BBMEFs depleted for WRN and WRNIP1 and treated with the Polθ inhibitor ART558. As expected from the role of Polθ in conducting error-prone TLS through CPDs, treatment with ART558 reduces mutation frequency in UV-irradiated BBMEFs near to that in unirradiated cells (*Table 1*). Importantly, in BBMEFs co-depleted for WRN and WRNIP1 treated with ART558 and expressing E84A WRN and K274A WRNIP1 or K577A WRN and K274A WRNIP1, UV-induced mutation frequency resulting from error-prone TLS through CPDs by Pol η rises to ~57 x 10$^{-5}$ (*Table 1*, last column; *Figure 2C*). These results concur with the inferences derived from mutational analyses in Polθ proficient cells for the additive effects of combinations of these WRN and WRNIP1 mutations on elevating the error-proneness of TLS through CPDs by Pol η (*Figure 2A*).

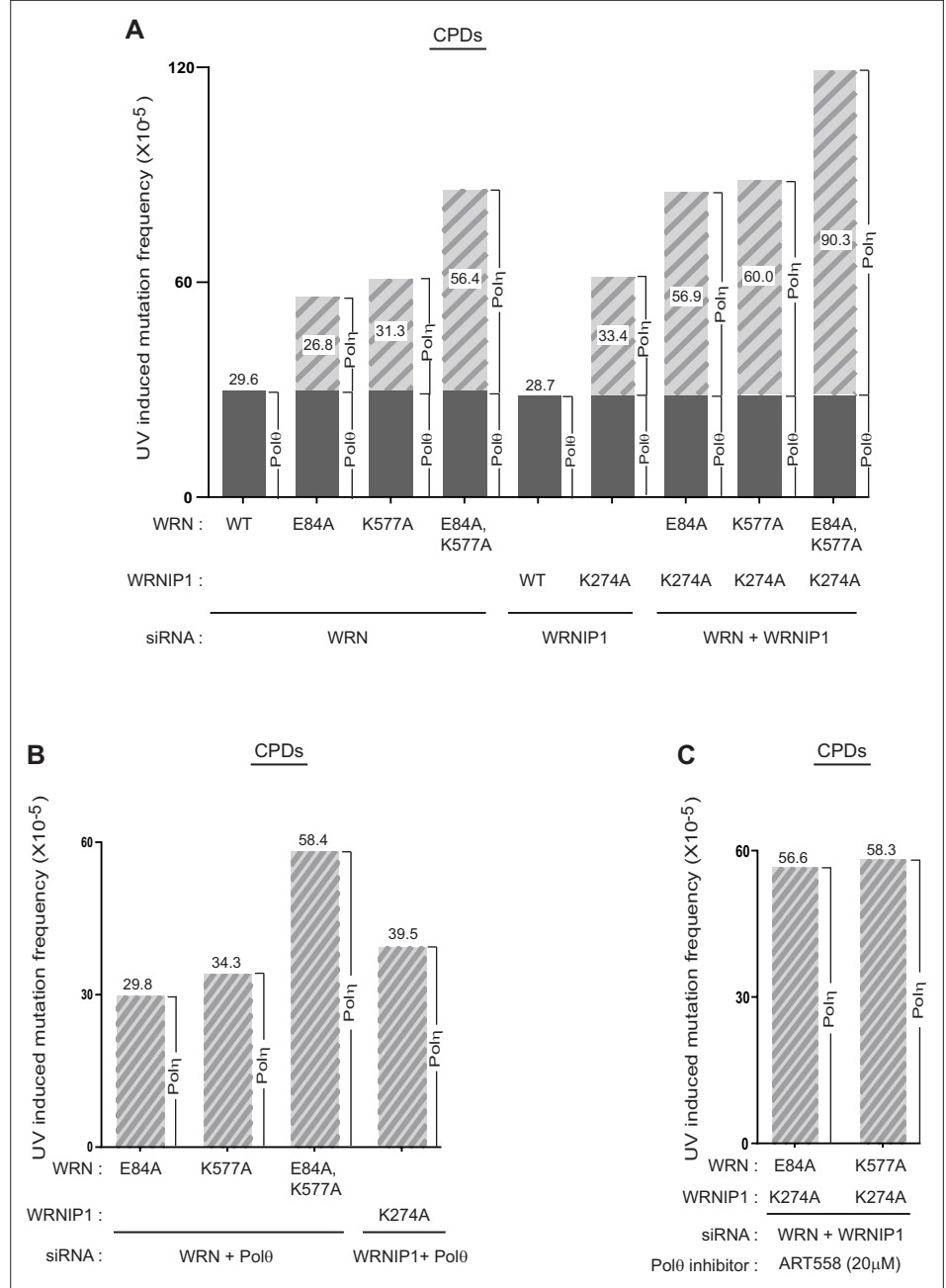

**Figure 2.** Imposition of error-proneness on Pol $\eta$ TLS through CPDs by defects in WRN ATPase, WRNIP1 ATPase, WRN 3'→5' exonuclease, or by their combinations. (**A**) UV-induced mutation frequencies resulting from TLS through CPDs by Pol $\eta$ in the *cII* gene in BBMEFs expressing E84A WRN, K577A WRN, K274A WRNIP1, or combinations of these mutant proteins. The contribution of Pol $\eta$ to UV-induced mutation frequency in BBMEFs defective in WRN ATPase, WRNIP1 ATPase, WRN exonuclease, or in combinations of these activities is indicated within the bar diagram. The simultaneous absence of WRN ATPase and exonuclease activities and WRNIP1 ATPase activity imposes a mutation frequency of ~90 x 10⁻⁵ upon error-free TLS through CPDs by Pol $\eta$ (A, last bar). (**B**) Verification that the increase in UV-induced mutation frequencies seen in the absence of WRN exonuclease, WRN ATPase, or WRNIP1 ATPase activities in (**A**) results from the error-proneness imposed upon Pol $\eta$ by the absence of these activities. (**C**) Verification that the increase in UV-induced mutation frequencies seen in the simultaneous absence of both WRN exonuclease and WRNIP1 ATPase activities or in the absence of both the WRN and WRNIP1 ATPase activities in (**A**) results from the error-proneness imposed upon Pol $\eta$ by the absence of these activities.

## Defects in WRN and WRNIP1 ATPase activities cause a diversity of nucleotide misincorporations opposite CPDs by Polη

UV-induced C>T mutations in the *cII* gene in WT BBMEFs resulting from error-prone TLS through CPDs by Polθ are clustered at hot spots at 11 dipyrimidine sites. In BBMEFs depleted for Polθ and expressing the exonuclease-defective E84A WRN, C>T and CC >TT tandem mutations resulting from error-prone TLS by Pol $\eta$ occur mostly at the same hot spots as those generated by Polθ (*Yoon et al., 2024*). The very high prevalence of C>T or CC >TT mutations to the almost exclusion of all other mutational changes in cells deficient in WRN 3'→5' exonuclease activity could arise because Pol $\eta$ misinserts only an A opposite the C residue of the CPD in CC, CT, or TC dipyrimidine sequences. Alternatively, it could derive from a role of WRN and WRNIP1 ATPase activities in preventing the diversity of nt misincorporations opposite CPDs by Pol $\eta$ .

To determine whether defects in WRN ATPase activity increase the variety of nt misincorporations opposite CPDs by Pol $\eta$ , we analyzed the spectrum of UV-induced mutations resulting from Pol $\eta$ 's role in TLS opposite CPDs in the *cII* gene in BBMEFs co-depleted for WRN and Polθ and expressing ATPase-defective K577A WRN (*Figure 3A*). Interestingly, both the pattern and the variety of nt misincorporations in BBMEFs expressing K577A WRN (*Figure 3A*) differ markedly from that in cells expressing the E84A WRN mutation (*Yoon et al., 2024*). Thus, in K577A WRN cells, C>T mutational hot spots are prevalent at positions 1, 2, 5, 7, 8, 9, and 10, but not at positions 3, 4, 6, and 11. Additionally, a new major C>T hot spot appears at the CT sequence at site a and a minor C>T hot spot occurs at the CT sequence at site b (*Figure 3A*). Quite remarkably, a number of hot spots manifest at other sites which entail mutational changes other than C>T. At hot spots at sites c, d, e, and h, the observed G>C changes would occur by the insertion of a C opposite the C residue of the CPD in the opposite strand; additionally, the C>G changes at hot spots at sites f and g would occur by the insertion of a C opposite the C residue of the CPD formed at these CT sequences (*Figure 3A*). The infrequent occurrence of a G>T change at sites 1, e, and 8 would occur from the insertion of a T opposite the C residue of the CPD in the opposite strand. Thus, defects in WRN ATPase greatly increase the misinsertion of an A or a C opposite the C residue of CPD.

To determine whether defects in WRNIP1 ATPase activity increase the variety of nt misinsertions opposite CPDs by Pol $\eta$ , we analyzed the spectrum of mutational hot spots in BBMEFs co-depleted for WRNIP1 and Polθ and expressing the ATPase-defective WRNIP1 K274A mutation (*Figure 3A*). Interestingly, defects in WRNIP1 ATPase activity confer a greater diversity of nt misincorporations opposite CPDs by Pol $\eta$ than those conferred by WRN ATPase deficiency. In WRNIP1 K274A cells, C>T mutational hot spots occur at sites 2, 3, 4, 5, 7, 8, 9, and 11 but not at sites 1, 6, or 10. However, the mutational pattern at site 9 differs from the pattern at other sites in the diversity of substitutions; in addition to the G>A change that would occur from the insertion of an A opposite the C residue of the CPD in the opposite strand, G>C, G>T, and GG >AA substitutions occur at this site (*Figure 3A*). These changes would involve the misincorporation of a C or T opposite the C residue of the CPD in the opposite strand, or the insertion of an A opposite both the 3'C and 5'C residues of the CPD in the opposite strand accounting for the tandem GG >AA mutations. Additionally, in BBMEFs expressing K274A WRNIP1, C>T mutational changes occur at sites a', d' and e'. Furthermore, the C>G mutational hot spot at the CT sequence present at sites f and g would result from the insertion of a C opposite the C residue of the CPD (*Figure 3A*). Thus, defects in WRNIP1 ATPase activity increase the misinsertion of an A as well as of a C opposite the C residue of CPD.

In addition to increasing A and C misinsertions opposite the C residue of CPD, defects in WRNIP1 ATPase activity confer an increase in nt misincorporations opposite the T residue of a CPD. Thus, the A>C change at sites b', c', and f' would occur from the insertion of a C opposite the T residue of the CPD in the opposite strand (*Figure 3A*). At site g', the prominent T>C hot spot would involve the insertion of a G opposite the T residue of the CPD. At site h', the A>G change would occur from the insertion of a G opposite the T residue of the CPD in the opposite strand, and an A>T change would occur from the insertion of a T opposite the T residue of the CPD (*Figure 3A*). Thus, in the absence of WRNIP1 ATPase activity, the misinsertion of C or G opposite the T residue of the CPD is elevated.

Since Pol $\eta$ misinsertions that occur in K577A WRN or K274A WRNIP1 cells could be removed by the WRN 3'→5' exonuclease activity, to get a more complete view of the diversity of Pol $\eta$ misinsertions in these ATPase-defective mutants, we analyzed the spectrum of UV mutations resulting from TLS through CPDs by Pol $\eta$ in BBMEFs co-depleted for WRN and Polθ and expressing E84A,

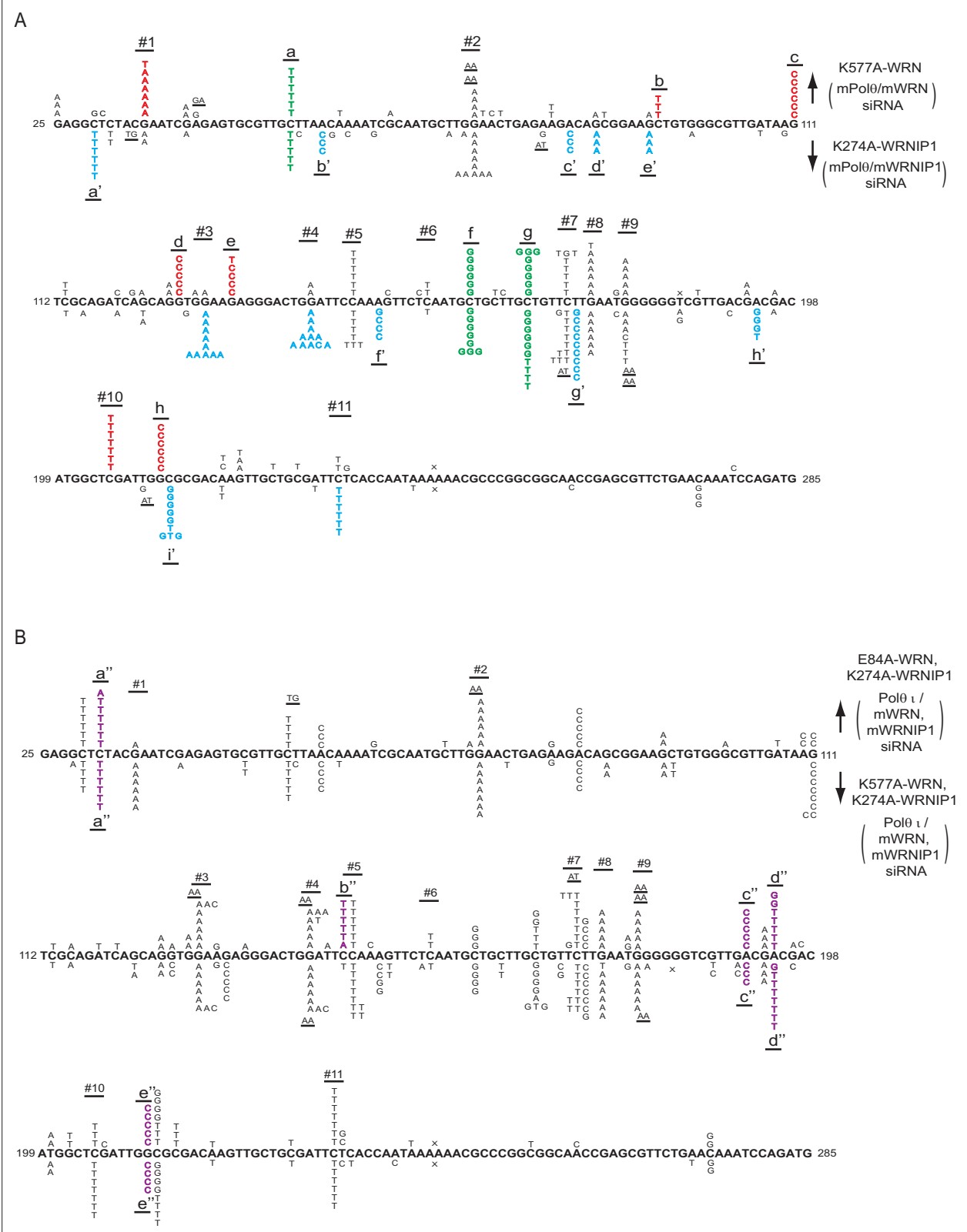

**Figure 3.** UV-induced (5 J/m²) mutational spectra resulting from TLS through CPDs by Pol η in the *cII* gene in BBMEFs expressing K577A WRN, K274A WRNIP1, E84A WRN K274A WRNIP1, or K577A WRN K274A WRNIP1. (**A**) Mutational spectra in BBMEFs co-depleted for WRN and Pol θ and expressing K577A WRN are shown above the sequence; mutational spectra in BBMEFs co-depleted for WRNP1 and Pol θ and expressing K274A WRNIP1 are shown below the sequence. Novel hot spots restricted to K577A WRN are indicated in red lettering and novel hot spots restricted to K274A WRNIP1

*Figure 3 continued on next page*

*Figure 3 continued*

are depicted in blue lettering. Green lettering indicates novel shared hot spots that appear in cells expressing either of these mutant proteins. (**B**) Mutational spectra in BBMEFs co-depleted for WRN and WRNIP1, treated with Pol θ inhibitor ART558 (Pol θ *i*), and expressing both E84A WRN and K274A WRNIP1 are shown above the sequence, and expressing both K577A WRN and K274A WRNIP1 are shown below the sequence. Novel hot spots that appear in BBMEFs expressing a combination of these mutant proteins are demarcated by violet lettering. The designations for the other mutational changes in (**A**) and (**B**) are: X, deletions; underlines, tandem mutations.

The online version of this article includes the following figure supplement(s) for figure 3:

**Figure supplement 1.** UV-induced mutational spectra resulting from TLS through CPDs by Pol η in the *cII* gene in BBMEFs expressing K577A WRN or E84A, K577A WRN.

K577A WRN (*Figure 3—figure supplement 1*) or in cells co-depleted for WRN and WRNIP1, treated with Polθ inhibitor ART558 and expressing E84A WRN, K274A WRNIP1 (*Figure 3B*). The mutational spectra in E84A, K577A WRN cells largely remain the same as in K577A WRN cells (*Figure 3—figure supplement 1*). In cells expressing E84A WRN together with K274A WRNIP1, however, the hot spot pattern differs in some aspects from that in K274A WRNIP1 cells. In addition to exhibiting the hot spot features of K274A WRNIP1, the mutational spectra in E84A WRN, K274A WRNIP1 exposes novel hot spots at sites <u>a"</u>, <u>b"</u>, <u>c"</u>, <u>d"</u>, and <u>e"</u> (*Figure 3B*). While the hot spots at <u>a"</u> and <u>b"</u> are C>T mutational changes, the hot spot at <u>c"</u> would result from the insertion of C opposite the T residue of CPD in the opposite strand. The hot spot at <u>d"</u> would result from the insertion of a T or a G opposite the T residue of CPD in the opposite strand; this identifies that defects in WRNIP1 ATPase activity also increase the misinsertion of T opposite the T residue of CPD. The hot spot at <u>e"</u> would result from the insertion of C opposite the C residue of CPD in the opposite strand.

To determine whether the absence of both WRN and WRNIP1 ATPase activities exposes additional novel features of nt misinsertions opposite CPDs by Pol η, we analyzed the mutational spectra in BBMEFs co-depleted for WRN and WRNIP1, treated with the Polθ inhibitor ART558, and expressing both the K577A WRN and K274A WRNIP1 (*Figure 3B*). In addition to the combined features of K577A WRN and K274A WRNIP1 mutational spectra, we observe novel hot spots at sites <u>a"</u>, <u>c"</u>, <u>d"</u>, and <u>e"</u> (*Figure 3B*). These hot spots are similar to those in BBMEFs expressing E84A WRN together with K274A WRNIP1 (*Figure 3B*). Since the A>C changes at <u>c"</u> would arise from the insertion of C opposite the T residue of CPD in the opposite strand, and A>T change at <u>d"</u> would result from the insertion of T opposite the T residue of a CPD in the opposite strand, the appearance of these hot spots in the absence of both the WRN and WRNIP1 ATPase activities would suggest a role of WRN ATPase activity in restraining the incorporation of a C or a T opposite the T residue of CPD in the absence of WRNIP1 ATPase activity.

Overall, these mutational data show that whereas defects in WRN ATPase activity primarily increase the misinsertion of an A or a C opposite the C residue of CPD by Pol η, defects in WRNIP1 ATPase activity, in addition to increasing the misinsertion of an A or a C opposite the C residue of CPD, increase the misinsertion of C, G, or T opposite the T residue of CPD by Pol η. Additionally, defects in these ATPase activities cause mutational hot spot formation in different sequence contexts.

## Defects in WRN ATPase, WRNIP1 ATPase, and WRN 3'→5' exonuclease impart an additive increase in the error-proneness of TLS opposite (6-4) photoproducts by Polη and Polι

TLS opposite (6-4) PPs is conducted *via* error-prone Pol η/Polθ or Pol ι/Polθ pathways in which following nt insertion opposite (6-4) PPs by Pol η or Pol ι, Polθ would extend synthesis (*Yoon et al., 2010b*; *Yoon et al., 2019b*). In the alternative pathway, error-free TLS through (6-4) PPs is conducted by Pol λ together with Pol ζ (*Yoon et al., 2021b*). Since WRN and WRNIP1 are required for TLS opposite (6-4) PPs by Pol η and Pol ι (*Yoon et al., 2024*), a defect in WRN ATPase, WRN exonuclease, or WRNIP1 ATPase would impact the fidelity of TLS by these Pols. To determine whether WRN and WRNIP1 ATPase activities modulate the fidelity of Pols η and ι for TLS opposite (6-4) PPs, we analyzed the effects of K577A WRN and K274A WRNIP1 on the frequency of UV-induced mutations in the *cII* gene in BBMEFs expressing CPD photolyase (*Figure 4A*). In this BBMEF cell line, spontaneous mutations occur at a frequency of ~15 x $10^{-5}$ in cells expressing WT WRN (*Figure 4A*). In UV-irradiated BBMEFs expressing WT WRN, mutations occur at a frequency of ~28 x $10^{-5}$. Thus, in WT

A

| siRNA[a] | Vector expressing | UV[b] | Photoreactivation | Mutation frequency[c] (x10^{-5}) | UV induced[d] mutation frequency (x10^{-5}) |
|---|---|---|---|---|---|
| WRN | Myc-WT-WRN | - | + | 14.6 ± 1.4 | - |
| WRN | Myc-WT-WRN | + | + | 28.3 ± 0.6 | 13.7 |
| WRN | Myc-E84A-WRN | + | + | 46.2 ± 1.0 | 31.6 |
| WRN | Myc-K577A-WRN | + | + | 47.2 ± 2.3 | 32.4 |
| WRN | Myc-E84A,K577A-WRN | + | + | 71.3 ± 0.9 | 56.7 |
| WRNIP1 | Flag-WT-WRNIP1 | + | + | 29.5 ± 1.1 | 14.9 |
| WRNIP1 | Flag-K274A-WRNIP1 | + | + | 50.7 ± 1.4 | 36.1 |
| WRN+ WRNIP1 | Myc-E84A-WRN + Flag-K274A-WRNIP1 | + | + | 70.4 ± 1.7 | 55.8 |
| WRN+ WRNIP1 | Myc-K577A-WRN + Flag-K274A-WRNIP1 | + | + | 70.6 ± 1.8 | 56.0 |
| WRN+ WRNIP1 | Myc-E84A,K577A-WRN+ Flag-K274A-WRNIP1 | + | + | 90.6 ± 0.9 | 76.0 |

**Figure 4.** UV-induced (5 J/m²) mutation frequencies resulting from TLS opposite (6-4) photoproducts in the *cII* gene in BBMEFs expressing E84A WRN, K577A WRN, K274A WRNIP1, or combinations of these mutant proteins. (**A**) UV mutations resulting from TLS opposite (6-4) PPs were examined in a BBMEF cell line expressing a CPD photolyase and photoreactivated with UVA (360 nm) light for 3 hr. Mutation frequencies and SEM were calculated from three to four independent experiments. UV-induced mutation frequency (last column) resulting from TLS through (6-4) PPs was calculated by subtracting the spontaneous mutation frequency (14.6x10⁻⁵) from the mutation frequency in UV irradiated cells. (**B**) Diagrammatic representation of elevation in error-proneness conferred by

*Figure 4 continued on next page*

*Figure 4 continued*

E84A WRN, K577A WRN, K274A WRNIP1, or by their combinations upon TLS opposite (6-4) PPs by Pol $\eta$ and Pol $\iota$ . The figure depicts the elevation in UV-induced mutation frequencies resulting from TLS opposite (6-4) PPs that occurs in BBMEFs expressing these WRN or WRNIP1 mutant proteins.

The online version of this article includes the following source data for figure 4:

**Source data 1.** Mutation frequencies from independent experiments for the data shown in *Figure 4A*.

cells, error-prone TLS through (6-4) PPs generates mutations at a frequency of ~14 x $10^{-5}$ (*Figure 4A*, last column). UV-induced mutation frequency rises to ~32 x $10^{-5}$ in cells expressing exonuclease-defective E84A WRN, and also in cells expressing ATPase-defective K577A WRN (*Figure 4A*, last column). In cells expressing E84A, K577A WRN, UV-induced mutation frequency rises to ~57 x $10^{-5}$ (*Figure 4A*). In cells expressing ATPase-defective K274A WRNIP1, UV-induced mutation frequency rises to ~36 x $10^{-5}$ and in cells expressing E84A WRN and K274A WRNIP1, or K577A WRN and K274A WRNIP1, UV-induced mutation frequency rises to ~56 x $10^{-5}$ (*Figure 4A*). In the absence of all three activities, UV-induced mutation frequency resulting from TLS through (6-4) PPs rises to ~76 x $10^{-5}$ (*Figure 4A*). Thus, defects in the WRN ATPase, WRNIP1 ATPase, or WRN exonuclease each increase the error-proneness of Pols $\eta$ and $\iota$ dependent TLS through (6-4) PPs to a similar extent, and the combination of these defects confers an additive increase in error-prone TLS such that in cells deficient in all three activities, UV-induced mutation frequency is elevated ~fivefold compared to that in WT cells (*Figure 4B*).

## Defects in WRN and WRNIP1 ATPase activities confer a distinct pattern of nucleotide misinsertions opposite (6-4) PPs in TLS mediated by Polη and Polι

In WT BBMEFs, TLS through (6-4) PPs by Pols $\eta$ and $\iota$ generates C>T mutational hot spots clustered at sites 1, 2, 3, 4, and 5 (*Yoon et al., 2010b*; *Yoon et al., 2021b*), and the pattern of mutations remains basically the same in WRN exonuclease deficient cells (*Yoon et al., 2024*). In BBMEFs expressing ATPase-deficient K577A WRN, in addition to hot spots at sites 1, 2, 3, and 5, new hot spots appear at sites a, b, c, and d (*Figure 5A*). While hot spots a, c, and d exhibit canonical C>T mutations in a potential dipyrimidine sequence, the G>C hot spot at site b would require the formation of UV photoproduct in a non-dipyrimidine sequence. This hot spot might result from TLS through the AC photoproduct in the opposite strand in which a C is inserted opposite the C residue of the photoproduct. The formation of such a UV photoproduct has been inferred from studies in yeast (*Laughery et al., 2020*). In BBMEFs expressing ATPase-deficient K274A WRNIP1, C>T mutational hot spots occur at sites 1 and 5 and at sites a and d; additional hot spots appear at sites a', b', and c'; wherein at site a', tandem AA misinsertions would occur opposite the CC residues in the opposite strand, and at sites b' and c', misinsertions would occur at UV photoproducts formed at non-dipyrimidine sequences (*Figure 5A*). These analyses indicate that defects in the WRN ATPase or WRNIP1 ATPase elevate the misinsertion of an A opposite the C residue of (6-4) PPs and also the misinsertions that occur at UV photoproducts that are presumably formed at non-dipyrimidine sequences.

Next, we determined whether the deficiency of both WRN and WRNIP1 ATPase activities generates a pattern of mutational hot spots different from that in cells deficient for either of these activities alone. However, the mutational pattern in the absence of both the WRN and WRNIP1 ATPases primarily exhibits features of both the deficiencies with the minor exception of a hot spot at position e" where the G>A or G>T change would involve the misinsertion of an A or a T opposite the 3'C of the CC sequence in the opposite strand (*Figure 5—figure supplement 1A*). The pattern of nt misincorporations in E84A, K577A WRN cells basically remains the same as in K577A WRN cells (*Figure 5—figure supplement 1B*).

Interestingly, the mutational spectra in E84A WRN, K274A WRNIP1 cells differ strikingly from that in K274A WRNIP1 cells (*Figure 5B*). In addition to exhibiting the K274A WRNIP1 features, E84A WRN, K274A WRNIP1 cells reveal novel hot spots at sites d', e', f', g', h', i', j', k', and l' (*Figure 5B*). These hot spots expose a variety of nt misincorporations that occur opposite (6-4) PPs in cells deficient in WRNIP1 ATPase activity but are removed by the WRN 3'→5' exonuclease activity; hence, they become evident in the absence of WRN's exonuclease activity. Among these hot spots, the T>C hot

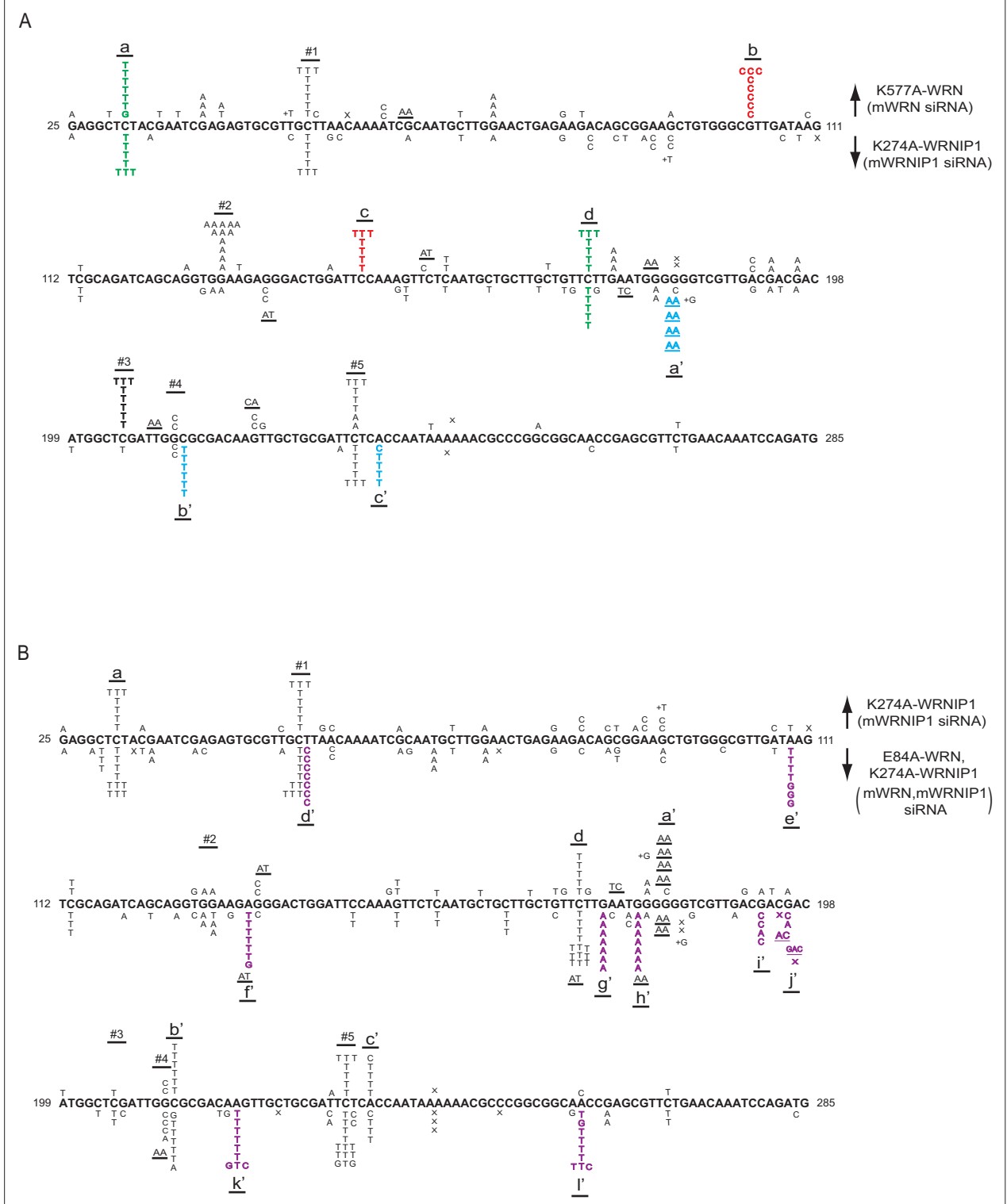

**Figure 5.** UV-induced (5 J/m²) mutational spectra resulting from TLS opposite (6-4) PPs by Pol η and Pol ι in the *cII* gene in BBMEFs expressing K577A WRN, K274A WRNIP1, or E84A WRN K274A WRNIP1. (**A**) Mutational spectra in BBMEFs depleted for WRN and expressing K577A WRN are shown above the sequence; mutational spectra in BBMEFs depleted for WRNIP1 and expressing K274A WRNIP1 are shown below the sequence. Novel hot spots restricted to K577A WRN are indicated by red lettering, and novel hot spots restricted to K274A WRNIP1 are indicated in blue lettering. Green lettering indicates novel shared hot spots that appear in cells expressing either of these mutant proteins. Hot spots 1 and 5 in WT cells are also present in cells expressing either of these mutant proteins. However, hot spots 2 and 3 present in WT cells are present only in cells expressing K577A WRN.

*Figure 5 continued on next page*

*Figure 5 continued*

(**B**) Mutational spectra in BBMEFs depleted for WRNIP1 and expressing K274A WRNIP1 are shown above the sequence; and mutational spectra in BBMEFs co-depleted for WRN and WRNIP1 and expressing E84A WRN and K274A WRNIP1 are shown below the sequence. Novel hot spots that appear in BBMEFs expressing E84A WRN, K274A WRNIP1 are indicated by violet lettering.

The online version of this article includes the following figure supplement(s) for figure 5:

**Figure supplement 1.** UV-induced mutational spectra resulting from TLS opposite (6-4) PPs by Pol $\eta$ and Pol $\iota$ in the *cII* gene in BBMEFs expressing both K577A WRN and K274A WRNIP1, or K577A WRN, or E84A, K577A WRN.

spot at position <u>d′</u> would result from insertion of G opposite the T residue of (6-4) PP; and at position <u>e′</u>, the A>T and A>G changes would occur by insertion of T or G opposite the T residue of (6-4) PP in the opposite strand. The A>T hot spots at sites <u>f′</u>, <u>k′</u>, and <u>l′</u> would involve T insertion opposite the T residue of (6-4) PP in the opposite strand. Additionally, the less frequent A>G and A>C changes at sites <u>k′</u> and <u>l′</u> would occur from insertion of G or C opposite the T residue of (6-4) PP in the opposite strand. And, whereas the novel G>A hot spots at sites <u>g′</u> and <u>h′</u> represent canonical C>T change, the G>C change at site <u>i′</u> would involve C insertion opposite the C residue of the dipyrimidine sequence in the opposite strand (***Figure 5B***).

Thus, overall, defects in WRN and WRNIP1 ATPase activities greatly elevate the misinsertion of an A opposite the C residue of (6-4) PP; additionally, defects in WRNIP1 ATPase activity engender a large increase in the misinsertion of T and, to a lesser extent, the misinsertion of G opposite the T residue of (6-4) PP. Furthermore, defects in WRN or WRNIP1 ATPase activities cause mutational hot spot formation in different sequence contexts, and they expose misinsertions at UV photoproducts that presumably form at non-dipyrimidine sites.

## Defects in WRNIP1 ATPase activity impair WRN 3′→5′ exonuclease function in the removal of Polκ misinsertions in TLS at the Tg lesion

TLS through the Tg lesion is conducted by error-free Polκ/Pol $\zeta$ pathway in which following nt insertion opposite Tg by Polκ, Pol $\zeta$ would extend synthesis (***Yoon et al., 2010a***). In the alternative pathway, Polθ promotes error-prone TLS and generates ~2% mutational products in which a wrong nt is inserted opposite Tg (***Yoon et al., 2014***; ***Table 2***). The frequency of TLS through the Tg adduct is not affected by the K274A WRNIP1, K577A WRN, or E84A WRN mutations or by their combinations (***Supplementary file 1A***). However, defects in WRNIP1 ATPase activity confer a high degree of error-proneness on TLS at the Tg lesion, as the expression of K274A WRNIP1 in WT HFs raises the mutation frequency to ~9% (***Table 2***), and this elevation derives primarily from mutations that occur

**Table 2.** Mutation frequencies and nucleotides inserted opposite a thymine glycol carried on the leading strand DNA template of a duplex plasmid in wild type human fibroblasts or WRN $^{-/-}$ fibroblasts and expressing WRN and/or WRNIP1 mutant proteins.

| HFs | siRNA | Vector expressing | Number of *Kan*+ blue colonies sequenced | A | G | C | T | Other* | Mutation frequency (%) |
|---|---|---|---|---|---|---|---|---|---|
| | WRNIP1 | WT-WRNIP1 | 96 (2)[†] | 94 | - | - | 2 | - | 2.1 |
| WT | WRNIP1 | K274A-WRNIP1 | 156(14) | 142 | - | - | 4 | 10 | 9.0 |
| | NC | Myc-WT-WRN[‡] | 192 (4) | 188 | 1 | - | 3 | - | 2.1 |
| | NC | Myc-E84A-WRN [‡] | 208 (15) | 193 | 2 | - | 4 | 9 | 7.2 |
| | NC | Myc-K577A-WRN | 160 (3) | 157 | - | - | 3 | - | 1.9 |
| | NC | Myc-E84A,K577A-WRN | 140 (11) | 129 | 2 | - | 1 | 8 | 7.9 |
| | WRNIP1 | K577A-WRN+K274A-WRNIP1 | 90 (7) | 94 | - | - | 1 | 6 | 7.8 |
| WRN $^{-/-}$ | WRNIP1 | E84A-WRN+K274A-WRNIP1 | 184(16) | 168 | 2 | - | 3 | 11 | 8.7 |

*Mutations occurred at the 5′ template residue next to Tg lesion. The sequence 5′-CAA<u>T</u>TgG-3′ is changed to 5′-CAA<u>C</u>TG-3′. The corresponding 5′ residues are underlined.

[†]Numbers of colonies where TLS occurred by insertion of a nucleotide other than an A are shown in parenthesis.

[‡]These data have been published previously (***Yoon et al., 2024***) and are shown here for comparison.

from insertion of G opposite the next T on the 5' side of the Tg lesion, resulting in 5' T Tg >5' CT change (*Table 2*). Since Polκ conducts error-free TLS at the Tg lesion and since WRN and WRNIP1 are required for TLS by Polκ (*Yoon et al., 2024*), this increase in mutation frequency would accrue from the error-proneness imposed upon Polκ by the lack of WRNIP1 ATPase activity.

By contrast, ATPase-defective K577A WRN has no adverse effect on error-free TLS by Polκ as the mutation frequency and the mutational pattern remain the same in WRN$^{-/-}$ HFs expressing K577A WRN as in WT WRN and these result from Polθ errors; moreover, the frequency of error-prone TLS and the mutational pattern in WRN$^{-/-}$ HFs expressing E84A K577A WRN remain the same as in E84A WRN (*Table 2*). The lack of any role of WRN ATPase activity in the fidelity of TLS by Polκ was further confirmed from the observation that the frequency and mutational pattern in WRN$^{-/-}$ HFs expressing K577A WRN and K274A WRNIP1 together remain the same as in K274A WRNIP1 (*Table 2*). To determine whether the WRNIP1 ATPase and WRN exonuclease activities act independently, we analyzed the frequency and pattern of mutational changes in WRN$^{-/-}$ HFs expressing both the E84A WRN and K274A WRNIP1 proteins together. Unexpectedly, we find that the frequency of mutations and the prevalence of T>C change at the 5'T next to Tg remain nearly the same in the absence of both these activities as in the absence of WRNIP1 ATPase or WRN 3'→5' exonuclease activity (*Table 2*). This epistatic interaction implicates a role of WRNIP1 ATPase in facilitating the action of WRN 3'→5' exonuclease in the removal of Polκ misinsertions at the Tg lesion.

## Defects in WRN and WRNIP1 ATPase activities elevate G misinsertions by Polι opposite εdA

TLS through εdA operates *via* an error-free Pol ι /Pol ζ dependent pathway or by an error-prone Polθ dependent pathway; additionally, a third pathway dependent upon Rev1 polymerase activity, although minor in its contribution to overall TLS, makes a significant contribution to error-prone TLS (*Yoon et al., 2019a*). TLS through the εdA adduct occurs at the same frequency in HFs defective in WRN ATPase or WRNIP1 ATPase activity, or in HFs defective in both these activities together with WRN 3'→5' exonuclease activity (*Supplementary file 1B*). As we have shown previously, in WRN$^{-/-}$ HFs expressing WT WRN, error-prone TLS by Polθ and Rev1 generates ~19% mutational TLS products and the exonuclease-deficient E84A WRN mutation elevates this mutation frequency to ~62% (*Yoon et al., 2024*; *Figure 6A*). This immense rise in mutation frequency accrues from the error-proneness that the absence of WRN exonuclease activity confers upon error-free TLS by Pol ι ; and it derives from highly elevated C misinsertions and to a lesser extent A misinsertions by Pol ι . These results indicate that even with intact WRN and WRNIP1 ATPase activities, Pol ι operates in a very highly error-prone manner and that this vast error-proneness is annulled by the removal of all the Pol ι misinsertions by the WRN 3'→5' exonuclease activity.

To determine whether WRNIP1 and WRN ATPase activities contribute to the fidelity of Pol ι for TLS opposite εdA, we first analyzed the effects of ATPase-defective K274A WRNIP1 in WT HFs. As shown in *Figure 6A*, in HFs expressing WT WRNIP1, error-prone TLS by Polθ and Rev1 generates ~15% mutational TLS products, and this frequency rises to ~26% in WT HFs expressing K274A WRNIP1. This increase in mutation frequency in K274A WRNIP1 cells accrues primarily from an elevation in G misinsertions to ~9% from ~1% in HFs expressing WT WRNIP1. In WRN$^{-/-}$ HFs expressing ATPase-defective K577A WRN, the mutation frequency, the frequency of G misinsertions, and the pattern of misinsertions remain the same as in WRN$^{-/-}$ HFs expressing WT WRN (*Figure 6A*). However, in WRN$^{-/-}$ HFs expressing E84A, K577A WRN, G misinsertions occur at a frequency of ~10% (*Figure 6A*), whereas G misinsertions occur at near WT levels in WRN$^{-/-}$ HFs expressing E84A or K577A WRN. This result suggests that defects in WRN ATPase activity elevate the frequency of G misinsertions by Pol ι (*Figure 6A*), but these misinsertions are largely removed by the WRN exonuclease activity. The elevation in G misinsertions in the absence of both ATPase activities is further supported by the evidence that in WRN$^{-/-}$ HFs expressing E84A, K577A WRN together with K274A WRNIP1, G misinsertions increase to ~20% (*Figure 6A and B*). To verify that this elevation in G misinsertion frequency accrues from the role of Pol ι and not of Rev1 in TLS, we determined the mutation frequency and pattern of nt misinsertions in WRN$^{-/-}$ HFs depleted for Pol ι and expressing E84A, K577A WRN together with K274A WRNIP1. Our result that the frequency of G misinsertions declines to ~1% (*Figure 6A and B*) confirms that defects in WRN and WRNIP1 ATPase activities confer an elevation in G misinsertions by Pol ι opposite εdA.

A

| HFs | siRNA | Vector expressing | No. of *Kan+* blue colonies sequenced | Nucleotide inserted | | | | Mutation frequency (%) |
|---|---|---|---|---|---|---|---|---|
| | | | | A | G | C | T | |
| WT | WRNIP1 | WT-WRNIP1 | 164 (25)[a] | 3 | 2 | 20 | 139 | 15.2 |
| | WRNIP1 | K274A-WRNIP1 | 124 (32) | 6 | 11 | 15 | 92 | 25.8 |
| WRN [-/-] | NC | Myc-WT-WRN[b] | 208 (40) | 8 | 4 | 28 | 168 | 19.2 |
| | NC | Myc-E84A-WRN[b] | 224 (138) | 40 | 8 | 90 | 86 | 61.6 |
| | NC | Myc-K577A-WRN | 96 (20) | 3 | 2 | 15 | 76 | 20.8 |
| | NC | Myc-E84A,K577A-WRN | 150 (96) | 14 | 15 | 67 | 54 | 64.0 |
| | WRNIP1 | E84A,K577A-WRN+ K274A-WRNIP1 | 128 (85) | 12 | 26 | 47 | 43 | 66.4 |
| | WRNIP1+ Polι | E84A,K577A-WRN+ K274A-WRNIP1 | 100 (24) | 6 | 1 | 17 | 76 | 24.0 |

B

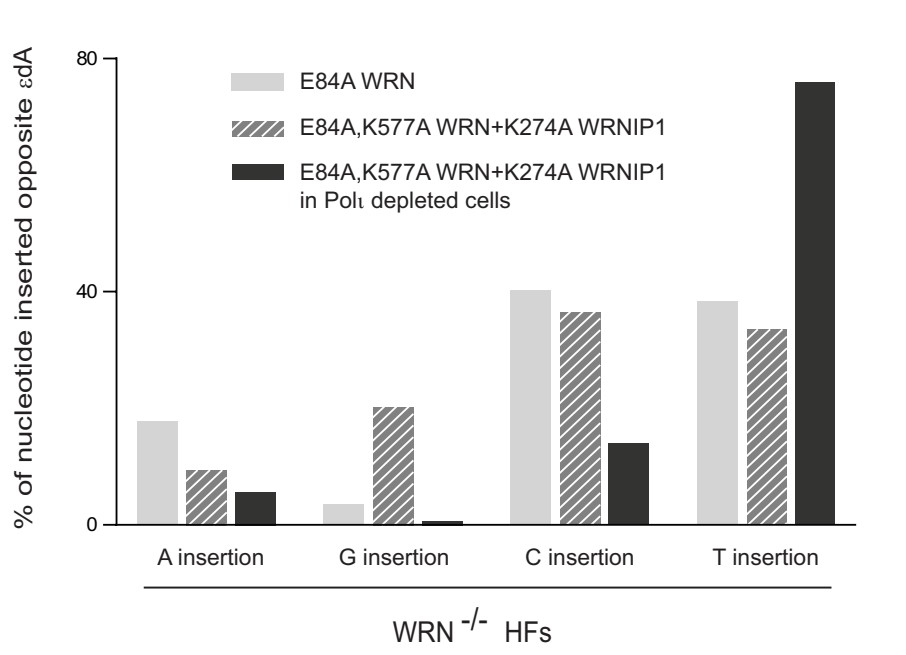

**Figure 6.** Defects in WRN and WRNIP1 ATPase activities elevate G misinsertions opposite εdA by Pol ι . (**A**) Effects of K274A WRNIP1, E84A WRN, K577A WRN mutations and their combinations on the error-proneness of TLS opposite εdA by Pol ι . Mutation frequencies and nts inserted opposite εdA carried on the leading strand template of a duplex plasmid in WT HFs or WRN[-/-] HFs expressing WRN and/or WRNIP1 mutant proteins. [a], Numbers in parentheses show the total number of mutations. [b], These data have been published previously (***Yoon et al., 2024***) and are shown here

*Figure 6 continued on next page*

*Figure 6 continued*

for comparison. (**B**) Diagrammatic representation of A, G, C, or T insertions opposite εdA by Pol ι that occur in WRN⁻ᐟ⁻ HFs expressing E84A WRN or expressing E84A, K577A WRN together with K274A WRNIP1.

## Discussion

### WRN and WRNIP1 ATPases restrain nucleotide misincorporations opposite DNA lesions by Y-Family Pols

Our evidence that error-free TLS through CPDs by Pol η manifests high mutagenicity in BBMEFs defective in WRN or WRNIP1 ATPase activity demarcates a role of both these activities in imposing high fidelity on TLS through CPDs by Pol η . From the mutational pattern incurred by defects in WRN or WRNIP1 ATPase activity, we infer a role for both these activities in restraining the misincorporation of an A or a C opposite the C residue of the CPD by Pol η (*Figure 7A*). Furthermore, from the mutational pattern incurred by defects in WRNIP1 ATPase activity, we infer an additional role of WRNIP1 ATPase in preventing the misincorporation of a C, G, or a T opposite the T residue of CPD by Pol η (*Figure 7A*).

The additive increase in UV-induced mutation frequencies resulting from TLS through CPDs by Pol η in cells defective in WRN and WRNIP1 ATPase activities and in WRN exonuclease activity would ensue from the independent roles of WRN and WRNIP1 ATPase activities in restraining the variety of Pol η misinsertions and of WRN exonuclease in removing the misinserted nts. Even though with intact WRN and WRNIP1 ATPase activities, Pol η performs error-prone TLS opposite CPDs, the near absence of mutations other than C>T or CC >TT changes in cells deficient in WRN exonuclease activity (*Yoon et al., 2024*) indicates that the combined action of WRN and WRNIP1 ATPases prevents almost all the other Pol η misinsertions except for the misincorporation of an A opposite the C residue of CPDs. Removal of these misinsertions by the WRN exonuclease renders Pol η TLS opposite CPDs error-free (*Figure 7A*).

The additive increase in the frequency of UV-induced mutations generated from TLS opposite (6-4) PPs by Pol η and Pol ι in cells defective in both the WRN and WRNIP1 ATPase activities adds further evidence for the independent roles of WRN and WRNIP1 ATPase activities in elevating the fidelity of TLS by Y-family Pols. And the preponderance of C>T hot spots in BBMEFs expressing K577A WRN or K274A WRNIP1 conforms with a role for both ATPases in restraining misincorporation of an A opposite the C residue of the photoproduct; additionally, WRNIP1 ATPase prevents misincorporation of a T or a G opposite the T residue of (6-4) PP (*Figure 7B*). Altogether,the combined actions of WRN and WRNIP1 ATPases in preventing nt misincorporations opposite (6-4) PPs by Pol η and Pol ι and of the WRN exonuclease in expunging misincorporated nts confer an immense rise in the fidelity of these Pols for TLS opposite (6-4) PPs (*Figure 4*).

### Imposition of a tight configuration on the active site of Y-family Pols by the WRN and WRNIP1 ATPase activities

The role of DNA helicases in the opening of the DNA helix in DNA replication, transcription, and recombination is well established (*Chen et al., 2008*; *Aibara et al., 2021*; *Schilbach et al., 2021*; *Lewis et al., 2022*). In keeping with such roles, WRN helicase has been shown to unwind a variety of secondary DNA structures (*Chu and Hickson, 2009*). Our results show that inactivation of the WRN ATPase activity has no adverse effect on the replication of UV-damaged DNA but greatly enhances nt misinsertions opposite UV lesions by Y-family Pols, identifying a role for this activity in restraining nt misincorporation by Y-family Pols. It is difficult to explain how the WRN DNA unwinding activity could modulate TLS Pol fidelity. Hence, we suggest that rather than employing the WRN ATPase activity for unwinding the DNA helix, the role of this activity gets modified by the components of the Y-family Pol multiprotein ensemble – that include WRN, WRNIP1, Rev1, and likely other proteins – to impact the active site of the TLS Pol, limiting misincorporations. In the Y-family Pol ensemble, WRNIP1 DNA-dependent ATPase activity could also be modified to act on the active site of the TLS Pol. Regardless of the mechanisms involved, these studies identify an unprecedented role of WRN and WRNIP1 ATPase activities in imposing high fidelity on TLS by Y-family Pols – which play a pre-eminent role in promoting replication through a large variety of DNA lesions.

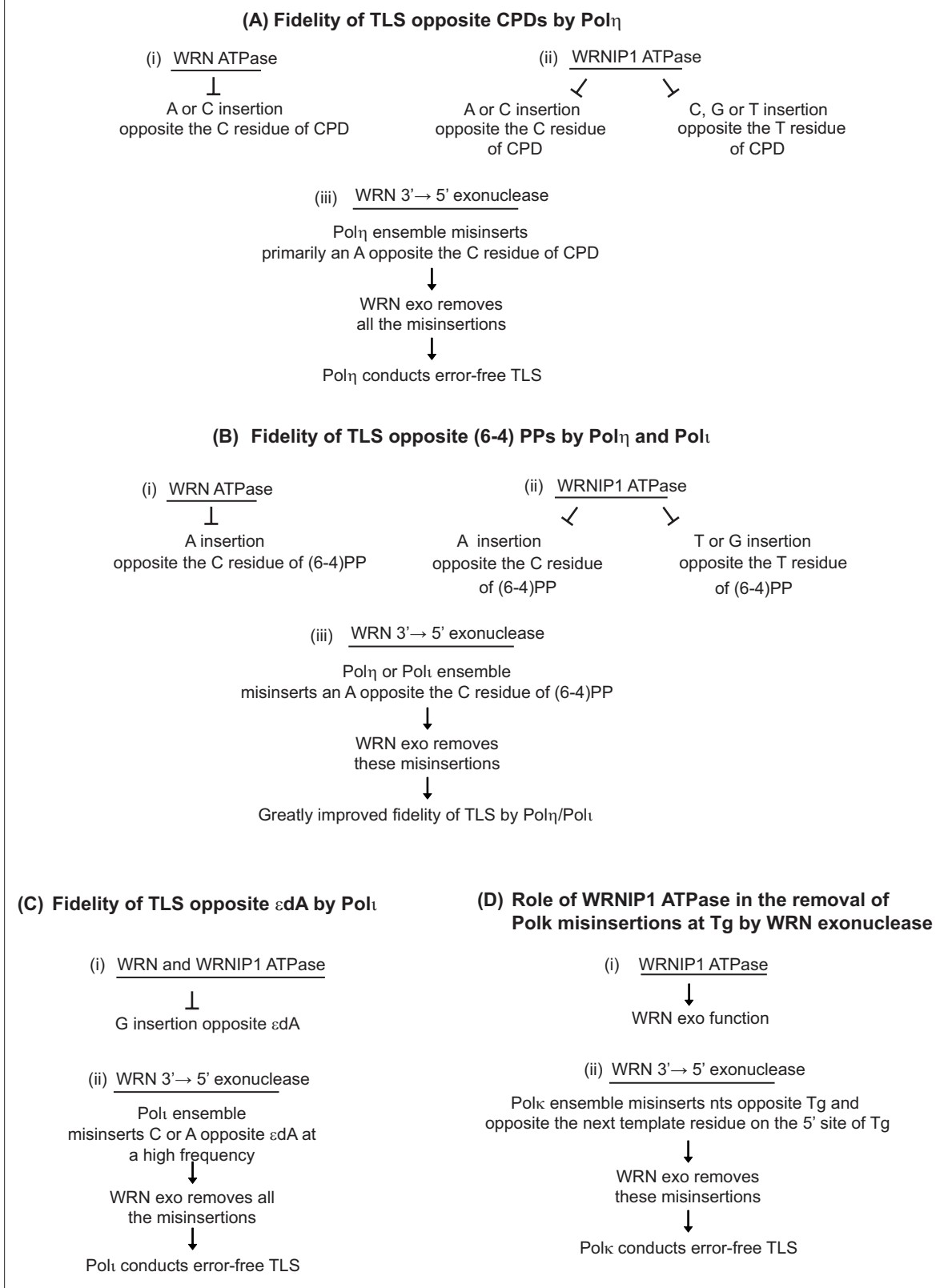

**Figure 7.** Roles of WRN ATPase, WRNIP1 ATPase, and WRN 3'→5' exonuclease activities in the high fidelity of TLS by Y-family Pols. (**A**) (i and ii) WRN and WRNIP1 ATPases restrain nt misincorporations by Pol η opposite CPDs; (iii) WRN exonuclease removes nt misinsertions opposite CPDs by the Pol η multiprotein ensemble. (**B**) (i and ii) WRN and WRNIP1 ATPases restrain nt misincorporations by Pol η or Pol ι opposite (6-4) PPs. (iii) WRN exonuclease removes nt misinsertions opposite (6-4) PPs by the Pol η or Pol ι multiprotein ensemble. (**C**) (i) WRN and WRNIP1 ATPases restrain G misinsertions

*Figure 7 continued on next page*

*Figure 7 continued*

by Pol $\iota$ opposite εdA. (ii) WRN exonuclease removes nt misinsertions opposite εdA by the Pol $\iota$ multiprotein ensemble. (**D**) (i) WRNIP1 ATPase promotes (↓) WRN exonuclease function in the removal of Polκ misinsertions at the Tg lesion. (ii) WRN exonuclease removes nt misinsertions by the Polκ multiprotein ensemble at the Tg lesion.

## Activation of the WRN 3'→5' exonuclease function by the WRNIP1 ATPase activity

In the replicative B-family Pols, the close proximity of the polymerase and exonuclease active sites in the same protein allows for the switching of a mismatched primer terminus from the polymerase active site to the exonuclease active site for the removal of misinserted nt (*Hogg et al., 2007*; *Darmawan et al., 2015*; *Jain et al., 2019*). In the Y-family Pol ensemble, the switching of a mismatched primer from the TLS Pol active site to the WRN exonuclease active site would require coordination of these two active sites, likely aided by their placement in close proximity to one another. The lack of requirement of WRN or WRNIP1 ATPase activity for the WRN exonuclease function in the removal of nts misinserted by Pol $\eta$ opposite CPDs or of nts misinserted by Pol $\eta$ or Pol $\iota$ opposite (6-4) PPs would suggest that opposite these DNA lesions, the coordination of the TLS Pol active site with the WRN exonuclease active site for the removal of misinserted nts is attained in the TLS Pol ensemble without the requirement of WRN or WRNIP1 ATPase activities. By contrast, the requirement of WRNIP1 ATPase activity for WRN exonuclease function in the removal of Polκ misinsertions at the Tg lesion might suggest that nt misinsertion at this site alters the configuration of the Polκ active site such that the proximity of the Polκ active site to the WRN exonuclease active site is disrupted and WRNIP1 ATPase activity aids in restoring that proximity. Regardless of the mechanism(s) involved, the requirement of WRNIP1 ATPase activity for the WRN exonuclease function in the removal of Polκ misinsertions implicates a close coordination in the actions of these different components of the Y-family Pol multiprotein ensemble.

## Imposition of high fidelity on TLS by Y-family Pols by the combined action of WRN and WRNIP1 ATPase activities and WRN 3'→5' exonuclease activity

Altogether, our studies show that the combined actions of WRN and WRNIP1 ATPases and WRN 3'→5' exonuclease confer such a high elevation in the fidelity of TLS by Y-family Pols that these very highly error-prone TLS Pols conduct TLS in an entirely or predominantly error-free manner. The relative contributions of these activities to the fidelity of TLS Pols, however, vary, depending upon the DNA lesion and the TLS Pol involved. Thus, for error-free TLS through CPDs by Pol $\eta$, all three activities make a prominent contribution – the WRN and WRNIP1 ATPases by restraining nt misincorporations and the WRN exonuclease by expunging nts misincorporated by Pol $\eta$ (*Figure 7A*). In a similar manner, the combined actions of all three activities confer an immense rise in the fidelity of Pol $\eta$ and Pol $\iota$ for TLS opposite (6-4) PPs (*Figure 7B*). For error-free TLS opposite εdA by Pol $\iota$, however, the role of WRN exonuclease in the removal of misinserted nts makes a greater contribution than the WRN and WRNIP1 ATPase activities do by curtailing G misinsertions (*Figure 7C*). At the Tg lesion, WRNIP1 ATPase promotes the WRN exonuclease function in the removal of Polκ misinsertions (*Figure 7D*).

Our evidence that the combined action of WRN and WRNIP1 ATPases along with WRN 3' to 5' exonuclease confers an enormous rise in the fidelity of TLS by Y-family Pols identifies the means by which these otherwise highly error-prone TLS Pols have been adapted to function in an error-free manner; thereby, providing a safeguard against genomic instability and tumorigenesis, as opposed to the deeply set percept that they contribute to it.

## Materials and methods

### Cell lines and cell culture

WT human fibroblast (GM00637) and WRN-deficient AG11395 cell line (*Dhillon et al., 2007*) derived from AG00780G fibroblasts (*Saito and Moses, 1991*) were obtained from Corriell Institute Cell Repository. We refer to WRN-deficient cells as WRN$^{-/-}$. Cell lines were authenticated by STR analysis and

verified to be free of mycoplasma contamination. These and BBMEF cell lines (Agilent, Cat# 726010) and the methods for their growth have been described previously (*Yoon et al., 2024*).

## Construction of plasmid vectors containing aTg or an εdA lesion

The heteroduplex TLS vectors containing a thymine glycol, or an 1-N⁶-etheno dA on the leading strand template were constructed as described previously (*Yoon et al., 2010a*; *Yoon et al., 2019a*).

## Translesion synthesis assays in human cells

For siRNA knockdown of WRNIP1, HPLC-purified duplex siRNA for human WRNIP1 was purchased from Thermo Fisher Scientific. The sense sequence of WRNIP1 siRNA target sequence is 5'-GAAA-CAUAGCAUAAGGUUU-3' and the efficiency of WRNIP1 knockdown was verified by western blot analysis (*Figure 1—figure supplement 1*). The siRNA knockdown efficiency of WRN or TLS Pols as well as the detailed methods for TLS assay and mutation analyses have been described previously (*Yoon et al., 2009*; *Yoon et al., 2010a*; *Yoon et al., 2019a*; *Yoon et al., 2024*).

## Western blot analysis

48 hr after siRNA transfection, cells were lysed with RIPA buffer (1 x PBS, 1% IP-40, 0.5% sodium deoxycholate, 0.1% SDS). After 1 hr incubation on ice, cellular mixture was centrifuged and the supernatant was collected. Equivalent amounts (approximately 30 µg) of prepared cellular extracts were separated on a 10% SDS-polyacrylamide gel and transferred to a PVDF membrane (Bio-Rad). The membranes were probed with rabbit polyclonal WRN antibody (Novus Bio, Cat#: NB100-471), rabbit polyclonal WRNIP1 antibody (Novus Bio, Cat#: NBP2-38190), mouse monoclonal flag antibody (Sigma, Cat#: F1804) or mouse monoclonal myc antibody (Santa Cruz Biotechnology, Cat#: sc-40) for 1 hr. After washing with PBS buffer, the membranes were mixed with appropriate secondary antibodies conjugated with horseradish peroxidase. The signals were detected using ECL-Plus (GenDEPOT). For the loading control, anti-β-tubulin antibody (Cell Signaling, Cat#: 2146), or anti-LaminB1 antibody (Abcam, Cat#: ab16048) was used.

## Foci formation assay

HFs (GM637) stably expressing flag-wild type WRNIP1 or flag-K274A WRNIP1 were treated with siRNA and cultured on a coverslip with 50% confluence. After 48 hr incubation, cells were treated with UVC (30 J/m²). WRN⁻/⁻ HFs stably expressing myc-wild type WRN or myc-K577A WRN were cultured on a coverslip with 50% confluence. After 16 hr incubation, cells were treated with UVC (30 J/m²). After UV irradiation, fresh DMEM growth media were added and cells were incubated for 3 hr. After washing with PBS buffer, cells were pre-extracted in 0.2% Triton X-100 for 2 min and fixed with 4% paraformaldehyde for 20 min. Primary antibodies, mouse monoclonal flag antibody (Sigma, Cat#: F1804) or rabbit polyclonal WRN antibody (Novus Bio, Cat#: NB100-471) were diluted in blocking buffer and incubated for 1 hr followed by washing with PBS buffer. Secondary antibodies, goat anti-mouse Alexa 488 (Thermo Fisher Scientific, Cat# A-11001) or goat anti-rabbit Alexa 488 (Thermo Fisher Scientific, Cat# A-11034) were applied for 30 min. Nuclear staining was performed with DAPI (Thermo Fisher Scientific) in PBS buffer for 20 min. The fluorescent images were visualized and captured by fluorescence microscope (Nikon Eclipse 80i).

## DNA fiber assay

WRN⁻/⁻ HFs stably expressing myc vector control, myc-wild type WRN, myc-K577A WRN, myc-K577A WRN and flag-K274A WRNIP1, or myc-E84A, K577A-WRN and flag-K274A WRNIP1 were treated with siRNA. After 48 hr incubation, cells were pulse-labeled with 25 µM IdU (Sigma) for 20 min. Cells were then washed with PBS buffer twice and irradiated with UVC (10 J/m²). After UV irradiation, cells were labeled with 250 µM CldU for 20 min. DNA fibers were spread on glass slides, and slides were incubated in 2.5 M HCl for 90 min and then washed with PBS buffer. The slides were incubated in blocking buffer (5% BSA in PBS buffer) for 2 hr. Primary antibodies, rat anti-BrdU antibody (Abcam Cat#: Ab6326) and mouse anti-BrdU antibody (BD Bioscience Cat#: 347580) were diluted in blocking buffer and incubated for 1 hr followed by washing with PBS buffer. Secondary antibodies, goat anti-rat Alexa 594 (Thermo Fisher Scientific, Cat# A-11007) and goat anti-mouse Alexa 488 (Thermo

Fisher Scientific, Cat# A-11001) were applied for 30 min and slides were mounted with antifade gold mounting media (Invitrogen). Fibers were analyzed by Nikon Eclipse fluorescence microscope.

### Big blue transgenic mouse cell line and siRNA knockdown

The big blue transgenic mouse embryonic fibroblasts (BBMEFs) were grown in DMEM medium containing 10% FBS (GenDEPOT) and antibiotics. Duplex siRNA for mouse WRNIP1 was purchased from Santa Cruz Biotechnology. The efficiency of its knockdown was verified by western blot analysis (*Figure 1—figure supplement 1*). For the *cII* mutation assay, cells were plated on 10 cm plates at 50% confluence and 500 pmoles of duplex siRNAs were transfected using 50 µl of iMfectin transfection reagent (GenDEPOT) following the manufacturer's instructions.

### Stable expression of myc-wild type WRN, myc-E84A WRN, myc-K577A WRN, myc-E84A,K577A WRN or flag-wild type WRNIP1, flag-K274A WRNIP1 or combinations of these mutant proteins in WRN$^{-/-}$ HFs or BBMEFs

Plasmids containing myc-wild type-WRN, myc-E84A-WRN, myc-K577A-WRN or myc-E84A, K577A-WRN were transfected into WRN$^{-/-}$ (AG11395) HFs or BBMEFs by iMfectin transfection reagent (GenDEPOT). After 24 hr incubation, 2 µg/mL of puromycin (Thermo Fisher Scientific) were added to the culture media. After 3 days of incubation, cells were washed with PBS buffer and were continuously cultured with the media containing 1 µg/mL of puromycin for 2 weeks. Plasmids containing flag-wild type-WRN or flag-K274A-WRNIP1 were transfected into normal human fibroblasts (GM637) or BBMEFs. After 24 hr incubation, 50 µg/mL of zeocin (Thermo Fisher Scientific) were added to the culture media. After 3 days of incubation, cells were washed with PBS buffer and were continuously cultured with the media containing 25 µg/mL of zeocin for 2 weeks. For co-expression of WRN mutants and flag-K274A-WRNIP1, plasmids containing flag-K274A-WRNIP1 were transfected in WRN$^{-/-}$ HFs or BBMEFs stably expressing myc-E84A-WRN, myc-K577A-WRN, or myc-E84A,K577A-WRN by iMfectin transfection reagent. Cells were continuously cultured with the media containing 1 µg/mL of puromycin and 25 µg/mL of zeocin. Protein expressions were verified by western blot analysis (*Figure 1—figure supplement 1*).

### UV irradiation, photoreactivation, and cII mutational assays in BBMEFs

48 hr after siRNA knockdown, cells were washed with HBSS buffer (Invitrogen) and irradiated at 5 J/m$^2$ with UVC light, followed by photoreactivation for 3 hr at room temperature as previously described (*Yoon et al., 2009*; *Yoon et al., 2010b*). Fresh DMEM growth medium was then added and cells were incubated for 24 hr. After the 24 hr incubation, the second siRNA transfection was carried out to maintain the siRNA knockdown of the target gene(s). Cells were incubated for an additional 3 days to allow for mutation fixation. For Polθ inhibition, 24 hr after siRNA treatment, cells were treated with 20 µM ART558 (Med Chem Express) for 16 hr. Then, cells were washed with HBSS buffer (Invitrogen) and irradiated at 5 J/m$^2$ with UVC light followed by photoreactivation. Fresh DMEM growth containing 20 µM ART558 was then added and cells were incubated for 24 hr. Following that, the second siRNA transfection was carried out to maintain the siRNA knockdown. Cells were incubated for an additional 3 days with DMEM growth media containing 20 µM ART558. The genomic DNA was isolated using the genomic DNA isolation kit (QIAGEN). The LIZ shuttle vector was rescued from the genomic DNA by mixing DNA aliquots and transpack packaging extract (Stratagene), and the *cII* assay was carried out as previously described (*Yoon et al., 2009*; *Yoon et al., 2010b*). The mutation frequency was calculated by dividing the number of mutant plaques by the number of total plaques. For mutation analysis, the sequence of PCR products of the *cII* gene from the mutant plaques was analyzed as described previously (*Yoon et al., 2009*; *Yoon et al., 2010b*).

## Acknowledgements

We thank Robert Johnson for reading the manuscript and for helpful suggestions. We are grateful to Raymond Monnat (University of Washington) for the plasmid carrying myc-WT WRN, myc- E84A WRN, or myc-K577A WRN.

## Additional information

### Funding

| Funder | Grant reference number | Author |
|---|---|---|
| National Institutes of Health | R35 GM148364 | Louise Prakash |

The funders had no role in study design, data collection and interpretation, or the decision to submit the work for publication.

### Author contributions

Jung Hoon Yoon, Formal analysis, Investigation, Methodology, Software, Validation; Karthi Sellamuthu, Investigation, Methodology, Validation; Louise Prakash, Conceptualization, Funding acquisition, Project administration, Resources, Supervision, Writing – original draft; Satya Prakash, Conceptualization, Supervision, Writing – original draft, Writing – review and editing

### Author ORCIDs

Jung Hoon Yoon  https://orcid.org/0000-0002-1624-3635
Karthi Sellamuthu  https://orcid.org/0000-0002-5703-9462
Louise Prakash  https://orcid.org/0000-0001-9143-6261
Satya Prakash  https://orcid.org/0000-0001-7228-1444

Reviewer #1 (Public review): https://doi.org/10.7554/eLife.106934.2.sa1
Reviewer #2 (Public review): https://doi.org/10.7554/eLife.106934.2.sa2
Reviewer #3 (Public review): https://doi.org/10.7554/eLife.106934.2.sa3
Author response https://doi.org/10.7554/eLife.106934.2.sa4

## Additional files

### Supplementary files

Supplementary file 1. TLS frequencies opposite a Tg (A) or an εdA lesion (B) in cells expressing WRN or WRNIP1 mutant proteins.

MDAR checklist

### Data availability

All data generated or analyzed during this study are included in the manuscript and supplementary materials.

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
