## [Editor Report · eLife Assessment]

This manuscript reports an **important** finding for understanding the molecular mechanisms of mutagenesis, carcinogenesis, and senescence. It follows a previous report showing that the Werner syndrome protein WRN and its interacting protein WRNIP1 are indispensable for translesion DNA synthesis (TLS) by Y-family DNA polymerases (Pols). The manuscript provides **convincing** evidence that WRN and WRNIP1 ATPases, in addition to the previously reported role of the WRN 3'>5' exonuclease activity, are essential for promoting the fidelity of replication through DNA lesions by Y-family Pols in human cells.

---

## [Referee Report · Reviewer #1 (Public review)]

Summary:

Y-family polymerases, such as polymerases eta, iota, and kappa, have low fidelity relative to other polymerases involved in DNA replication and repair. This is believed to be due to their active sites being less constrained than those of other polymerases. Paradoxically, work by this lab and others shows that in vivo, these Y-family polymerases are more error-free (less error-prone) during DNA damage bypass than would be expected given their low fidelity. For this reason, the authors have been focusing on other cellular factors that may increase the fidelity of Y-family polymerases. The current paper focuses on two such factors: WRN, which possesses exonuclease and helicase activities, and WRNIP1, which possesses a DNA-dependent ATPase.

Previously, this group showed that defects in the exonuclease function of WRN lead to a loss in the fidelity of polymerases eta and iota during DNA damage bypass, presumably by removing nucleotide misinsertions. The current paper extends this work by considering the ATPase activities of WRN and WRNIP1. The authors looked at the impact of various amino acid substitutions in these proteins on the fidelity of DNA damage bypass by Y-family polymerases. They did this by both measuring the mutation frequencies of these cell lines as well as the mutation spectra observed in them. They showed that the ATPase activities of both WRN and WRNIP1, as well as the exonuclease activities of WRN, are necessary high fidelity of Y-family polymerases in cells. They specifically examined the bypass of cyclobutene pyrimidine dimers by polymerase eta, the bypass of 6-4 photoproducts by polymerases eta and iota, and the bypass of ethenoadenine by polymerase iota. Moreover, they showed that WRNIP1 ATPase defects impair the WRN exonuclease from removing misinsertions by polymerase iota at thymine glycol lesions. These defects generally do not affect the efficiency of the bypass, only its fidelity.

Strengths:

The manuscript by Yoon et al is the latest in a series of important and impactful papers by this research group examining the cellular factors that enhance the fidelity of translesion synthesis by Y-family polymerases in human cell lines. Overall, the study is well designed, the data are clearly presented, and the conclusions are well supported and convincing. The authors also discuss a reasonable possibility that complex formation between the WRN and WRNIP1 proteins and Y-family polymerases could tighten the active sites of these polymerases to improve fidelity. Further studies are required to demonstrate this model, but it is a very exciting model that is well supported by the current data.

Weaknesses:

No weaknesses were identified by this reviewer.

---

## [Referee Report · Reviewer #2 (Public review)]

The authors of the present study are responsible for a previous study, which also showed that in response to DNA damage, Werner syndrome protein WRN, WRN interacting protein WRNIP1, and Rev1 assemble together with Y-family Pols (Polη, Polι, or Polκ), and that they are indispensable for Trans-Lesion-Synthesis (TLS) (Genes Dev 2024). They also identified a role of WRN's 3'→5' exonuclease activity in the high in vivo fidelity of TLS by Y-family, through UV-induced CPDs by Polη, through N6 ethenodeoxyadenosine (εdA) by Polι, through thymine glycol by Polκ, and through UV-induced (6-4) photoproducts by Polη and Polι. Thus, by removing nucleotides misinserted opposite DNA lesions by the Y-family Pols, WRN's 3'→5' exonuclease activity improves the fidelity of TLS by these Pols. The present work, which follows up on this previous work, reports the crucial role also of the ATPase activities of WRN and WRNIP1 in raising the fidelity of TLS by Y family Pols, in addition to the exonuclease activity, with an entirely different mechanism, which normally consists in unwinding of DNA containing secondary structures.

By using adequate cell line models and methodologies, notably DNA fiber, TLS, and mutation analyses assays, as well as specific ATPase point mutations, they found that progression of the replication forks through UV lesions was not affected in cells lacking the WRN exonuclease activity as well as the WRN and WRNIP1 ATPase activities, but occurs with a vast increase in error-prone TLS, notably through CPDs by Polη, with differential impacts on the nature of mutations between WRN ATPase and WRNIP1 ATPase. The relative contributions of these activities (exonuclease and ATPase) to the fidelity of TLS Pols, however, vary, depending upon the DNA lesion and the TLS Pol involved. Additionally, defects in these ATPase activities cause mutational hot spot formation in different sequence contexts. The authors provide evidence that the combined action of WRN and WRNIP1 ATPases, along with WRN 3' to 5' exonuclease, confers an enormous rise in the fidelity of TLS by Y-family Pols. They identify the means by which these otherwise highly error-prone TLS Pols have been adapted to function in an error-free manner. They suggest that WRNIP1 ATPases prevent misincorporations while WRN exonuclease removes misinserted nucleotides. This combination confers a vast increase in the fidelity of Y-family Pols, essential for genome stability.

Overall, this is a comprehensive and thoughtful manuscript, and all the findings reported are convincing and well supported. The data cannot be considered as entirely novel, as they follow-up on the recent 2024 publication by the same authors who unveiled that the exonuclease activity of WRN and WRNIP1 confers accuracy of TLS. The experimental methods are multiple and rigorous.

---

## [Referee Report · Reviewer #3 (Public review)]

Summary:

Replication through DNA lesions such as UV-induced pyrimidine dimers is mainly performed by Y-family pols. These translesion synthesis (TLS) pols are intrinsically error-prone. However, in living cells, TLS must be conducted in an error-free manner. This manuscript demonstrated that WRN and WRNIP1 ATPases play an important role in addition to WRN 3'>5' exonuclease in human cells.

Strengths:

The authors made use of WT human fibroblasts and WRN-deficient cell line for TLS assays in human cells and siRNA knock-down experiments to analyze TLS efficiency. For the cII mutation assay, the big blue mouse embryonic fibroblasts were used. These materials, as well as other Materials and Methods, had already been well established by this group or other groups. The authors used Pol eta, iota, kappa, and theta as TLS pols, and used UV-induced CPD, (6-4)PP, epsilon dA, and thymine glycol as DNA lesions. Thus, the authors examined the generality of their results in terms of TLS pols and DNA lesions.

Weaknesses:

Although the main part of this manuscript is the impact of the deficiencies of WRN and WRNIP1 ATPases on TLS by Y-family DNA polymerases, especially on TLS efficiency and mutation spectrum, many readers would be interested in how these ATPases could change molecular structure of Pol eta, because the structure of it have been studied for some time.

---

## [Author Response]

In the Weaknesses, Reviewer 3 suggests that in the Discussion, we comment upon whether WRN ATPase/3’-5’ helicase and WRNIP1 ATPase work on Y-family Pols additively or synergistically to raise fidelity. However, in the Discussion on page 20, we do comment on the role of WRN and WRNIP1 ATPase activities in conferring an additive increase in the fidelity of TLS by Y-family Pols.